# F-Box Genes in the Wheat Genome and Expression Profiling in Wheat at Different Developmental Stages

**DOI:** 10.3390/genes11101154

**Published:** 2020-09-30

**Authors:** Min Jeong Hong, Jin-Baek Kim, Yong Weon Seo, Dae Yeon Kim

**Affiliations:** 1Advanced Radiation Technology Institute, Korea Atomic Energy Research Institute, 29 Geumgu, Jeongeup 56212, Korea; hongmj@kaeri.re.kr (M.J.H.); jbkim74@kaeri.re.kr (J.-B.K.); 2Division of Biotechnology, Korea University, 145 Anam-ro, Seongbuk-Gu, Seoul 02841, Korea; seoag@korea.ac.kr; 3Institute of Animal Molecular Biotechnology, Korea University, 145 Anam-ro, Seongbuk-Gu, Seoul 02841, Korea

**Keywords:** developmental stage, F-box protein, wheat

## Abstract

Genes of the F-box family play specific roles in protein degradation by post-translational modification in several biological processes, including flowering, the regulation of circadian rhythms, photomorphogenesis, seed development, leaf senescence, and hormone signaling. F-box genes have not been previously investigated on a genome-wide scale; however, the establishment of the wheat (*Triticum aestivum* L.) reference genome sequence enabled a genome-based examination of the F-box genes to be conducted in the present study. In total, 1796 F-box genes were detected in the wheat genome and classified into various subgroups based on their functional C-terminal domain. The F-box genes were distributed among 21 chromosomes and most showed high sequence homology with F-box genes located on the homoeologous chromosomes because of allohexaploidy in the wheat genome. Additionally, a synteny analysis of wheat F-box genes was conducted in rice and *Brachypodium distachyon*. Transcriptome analysis during various wheat developmental stages and expression analysis by quantitative real-time PCR revealed that some F-box genes were specifically expressed in the vegetative and/or seed developmental stages. A genome-based examination and classification of F-box genes provide an opportunity to elucidate the biological functions of F-box genes in wheat.

## 1. Introduction

Biological and cellular processes in plants are regulated by several mechanisms, such as controlled gene expression, protein synthesis, protein modification, protein degradation, and interactions among molecules. The ubiquitin proteasome system (UPS), which selectively regulates protein degradation via the 26S proteasome, is a key mechanism for the post-translational control of several intracellular proteins. The UPS plays a significant role in the regulation of signal transduction, metabolic regulation, differentiation, cell cycle transition, and stress response by causing the degradation of specific proteins [1,2]. The UPS involves a cascade comprising three steps: the adenosine triphosphate-dependent activation of ubiquitin by ubiquitin-activating enzyme (E1), the transfer of ubiquitin to a conjugating enzyme (E2), and the conveyance of ubiquitin to a substrate protein by E3 [3]. Recently, hundreds of different E3s have been identified in the genomes of several plant species [4] and the number of E3s widely varies in the genomes of related species [5]. Several types of E3 ubiquitin ligases have been characterized by the presence of specific domains, including homology to the E6-AP C-terminus domain and a really interesting new gene/U-box domain, and Cullin–Ring ubiquitin ligase. RING/U-box E3 ligase can act as a single component and directly transfer ubiquitin to a target protein, whereas Cullin–RING ubiquitin ligases are multicomponents and work with the Skp1–Cullin–F-box (SCF) complex [6].

The SCF complex, which comprises the Skp1, Cullin, Rbx1, and F-box proteins, has been characterized in several species [7]. Cullin, the major structural scaffold for the SCF complex, connects Skp1 to Rbx1, and Skp1 binds F-box proteins via a substrate recognition function [8]. Numerous F-box proteins contain a conserved F-box (FBX) domain located in the 60 amino acids of the N-terminus, and can be classified by different C-terminal protein–protein interaction domains that recruit targets for proteasomal degradation. The C-terminal region of F-box proteins contains several protein–protein interaction domains, such as leucine-rich repeats, WD40, Kelch repeats, F-box-associated domain (FBA), or Tubby (Tub), which binds with target substrates and confers specificity to the F-box proteins [9].

Various members of the F-box gene family have been identified in the genomes of different plant species. At least 678, 913, 480, and 656 F-box proteins have been reported in *Oryza sativa*, *Medicago truncatula*, *Glycine max*, and *Arabidopsis thaliana*, respectively [10]. The presence of high numbers of F-box proteins in plant genomes suggests that numerous SCF complexes regulate various substrates and play specific regulatory roles through post-translational modification in several biological processes, such as the regulation of flowering and circadian rhythms [11], photomorphogenesis [12], leaf senescence [13], floral development [14], lateral shoot branching [15], gibberellic acid signaling [16], and auxin signaling [17]. Functional studies of F-box proteins in *Arabidopsis* are ongoing; however, little is known about the F-box proteins of wheat. In addition, the F-box genes of wheat have not been classified according to the C-terminal domain structures.

Wheat is allohexaploid (2*n* = 6x = 42; AABBDD) with a huge genome of approximately 17 Gbp that consists of three closely related homoeologous subgenomes comprising a high percentage of repetitive sequences and homoeologous DNA copies. A draft genome of wheat referred to as the International Wheat Genome Sequencing Consortium (IWGSC) chromosome survey sequence assembly (IWGSC 2014) was provided to the researcher [18], followed by the wheat assembly TGACv1, which provides chromosome arm-specific chromosome survey sequence reads [19]. More recently, the reference sequence of a bread wheat cultivar of Chinese Spring was made publicly available together with gene annotations [20]. In the current study, F-box genes were identified, selected, and classified based on a hidden Markov model (HMM) search of the wheat genome. Furthermore, a phylogenetic tree was constructed, and chromosomal locations were mapped for the 1796 genes that encode F-box proteins. The synteny between wheat F-box proteins and those of rice and *Brachypodium distachyon*, which was proposed as a model plant for monocotyledon species [21], was analyzed. Additionally, expression profiling the F-box genes of wheat was conducted at various developmental stages, including the vegetative and reproductive stages, using transcriptome data. The results of the present study provide novel information regarding the classification and expression of F-box proteins that may be useful for functional research on the different developmental stages of wheat.

## 2. Materials and Methods

### 2.1. Discovery of F-Box Genes by Sequence Analysis of the Wheat Genome

The IWGSC wheat reference sequence v1.0 (https://urgi.versailles.inra.fr/download/iwgsc/IWGSC_RefSeq_Annotations/v1.0/) was provided by Unité de Recherche Génomique Info (URGI, https://urgi.versailles.inra.fr/). HMM profiling of the F-box proteins was conducted with the HMM files of F-box (PF00646), F-box-like (PF12937), F-box-like2 (PF13013), FBA1 (PF07734), FBA2 (PF07735), FBA3 (PF08268), and FBD (PF08387) domains, which were provided by Pfam [22] and searched against a protein database of the wheat genome using the HMMER3 tool with default parameters [23]. Redundant F-box genes that did not contain an F-box domain in the C-terminal were removed, whereas F-box genes with a C-terminal FBX were verified using the SMART tool [24] and the Pfam protein database with an E-value threshold of 1 × 10^−5^ for significance. Additionally, the Blast2GO bioinformatics platform was used for the functional annotation of the F-box genes [25]. Local BlastX was conducted with peptide sequences of the *Poaceae* family sequences retrieved from the National Center for Biotechnology Information database to initiate gene ontology (GO) analysis for the construction of xml files. Next, mapping was performed to retrieve the gene names, GO terms, and protein sequences from the UniProt database, which was followed by annotation with default parameters (E-value < 1 × 10^-6^, annotation cutoff = 55, GO weight = 5).

### 2.2. Phylogenetic Analysis and Chromosomal Locations

For phylogenetic analysis, a phylogenetic tree file (ph file; Newick format) was created using the neighbor-joining method in ClustalW with the protein sequences of the selected wheat F-box genes. Bootstrapping was conducted with 1000 replications. The Interactive Tree of Life (iTOL) tool was applied to construct a phylogenetic tree of the F-box genes, and ClustalW was used to generate a tree file [26]. The chromosomal positions of the F-box genes were determined using information from the IWGSC Reference Sequence v1.0 and then plotted using MapChart 2.30 [27].

### 2.3. Synteny Analysis of F-Box Genes

The IWGSC Reference Sequence v1.0 provided by URGI, along with the *Brachypodium* and rice genome sequences, were downloaded from the Ensembl Plants database (http://plants.ensembl.org/) and used for a synteny analysis of the F-box genes. The BLASTP algorithm was used to predict the sequence homology of F-box genes to genes on homoeologous chromosomes in wheat, with an E-value threshold of <1 × 10^−10^, sequence identity of >90%, and bitscore of >500, as the homoeologous DNA copies from the three subgenomes (AA, BB, and DD) had high sequence similarity. For the synteny analysis of wheat against *Brachypodium* and rice, BLASTP was conducted with an E-value threshold of 1 × 10^−10^ and sequence identity of >80%. Intragenomic and intergenomic comparisons were conducted using Circos [28].

### 2.4. Plant Materials

The common wheat used in the present study was developed from a single plant descended from the F_10_ generation of a cross of Woori-mil (Korea RDA accession no. IT172221) × D-7 (an inbred line developed by Korea University, Fleming*4/3/PIO 2580//T83103 *2/Hamlet) as previously reported [29]. Seeds were vernalized at 4 °C for 4 weeks to synchronize growth and then transferred to an Incu Tissue plant culture vessel (72 × 72 × 22 mm^3^; SPL Life Sciences, Seongnam, Gyeonggi-do, Korea) containing a polypropylene net floating on Hoagland solution (Sigma-Aldrich, St. Louis, MO, USA). The plants were grown under controlled environments at 23–26 °C and a 16-h light/8-h dark cycle. Using the Zadoks growth scale (Z) [30] as a reference, three independent biological replicates of the following plant samples were collected at different developmental stages: leaves at Z13 (three leaves emerged; Stage 1), leaves at Z24 (main stem and four tillers present; Stage 2), leaves at Z51 (leaf at the tip of ear just visible, booting stage; Stage 3), spikelets at Z61 (beginning of anthesis; Stage 4), spikelets at Z73 (early milk development; Stage 5), spikelets at Z83 (early dough stage; Stage 6), and spikelets at Z91 (hard grain stage; Stage 7). Each collected sample was stored at −80 °C until analysis.

### 2.5. RNA Sequencing and Expression Profiling

Total RNA of leaves was extracted from Stage 1 to 4 using TRIzol reagent (Invitrogen, Waltham, MA, USA) and treated with DNase I to eliminate any contaminating genomic DNA. Total RNA extraction of developing seeds from Stage 5 to 7 was performed according to Meng et al. [31], for an effective inhibition of RNase activity and separation from the polysaccharides to extract maximum RNA solubility. RNA quality was assessed using the Agilent 2100 bioanalyzer (Agilent Technologies, Amstelveen, The Netherlands), and RNA quantification was performed using ND-2000 Spectrophotometer (Thermo Inc., Wilmington, DE, USA). Total RNA (10 μg) extracted from the samples was used to construct RNA sequencing (RNA-Seq) paired-end libraries with the TruSeq RNA Sample Preparation Kit (catalog #RS-122-2001; Illumina, San Diego, CA, USA). mRNA was isolated using the Poly(A) RNA Selection Kit (LEXOGEN, Inc., Vienna, Austria) and reverse transcribed into cDNA in accordance with the manufacturer’s instructions. The Agilent 2100 bioanalyzer was used to check the libraries, and the DNA High Sensitivity Kit was used to evaluate the mean fragment size. High-throughput sequencing was performed using the HiSeq 2000 platform (Illumina). The adaptor sequences were removed and sequence quality was tested using the BBduk tool (minimum length > 20, and Q > 20) before alignment. An index of the genome sequence was constructed and all reads were aligned to the wheat genome sequence using the HISAT2 alignment program with default parameters [32]. The HTSeqv0.6.1 high-throughput sequencing frame work was used to count the number of reads mapped to the exons of each gene [33]. The log_2_ transformed reads per kilobase of transcript per million mapped reads (RPKM) values were calculated and used to construct heatmaps of F-box gene expression at different developmental stages using R software. Mev software was used for k-means clustering of differentially expressed F-box genes [34]. For quantitative real-time (qRT)-PCR analysis, each RNA sample was treated with DNase I to completely digest contaminating genomic DNA, and first-strand cDNA was synthesized using the Power cDNA Synthesis kit (iNtRON Biotechnology, Seongnam, Gyeonggi-do, Korea) with 1 μg of total RNA and 2 × SYBR premix Ex *Taq* II (Takara, Shiga, Japan) in a 25 µL reaction volume using the iCycleriQ^TM^ Real-Time PCR System (Bio-Rad, Hercules, CA, USA). The sequences of the F-box proteins selected by k-means clustering methods were blasted against the wheat reference genome to compare genes with homoeologous sequences. Homoeologous regions specific to selected F-box genes were used to design gene-specific primers for the validation of the qRT-PCR products (Appendix A).

## 3. Results

### 3.1. Identification of F-Box Genes in the Wheat Genome

A total of 4363 putative F-box genes were identified by HMM profiling F-box proteins against the local protein database of the wheat genome using the HMMER3 tool. The putative F-box genes were re-examined to identify those with an FBX domain, whereas genes with redundant sequences were removed using the SMART tool and Pfam analysis. A total of 1796 F-box genes containing an FBX domain were selected for further study (Appendix A). The F-box genes were classified into 10 subgroups containing an FBX domain and other specific functional domains. Genes containing only one FBX domain in the N-terminal of the peptide sequence were the most abundant gene subgroup of the wheat genome (1370 genes). The groups of other F-box genes are referred to as FBA, FBD, DUF295, Kelch, Tub, PP2, Arm, and Cupin_8. Each of these subgroups contained FBX domains in the N-terminal and one or more known functional domains in the C-terminal of the peptide sequences. The FBA, FBD, DUF295, Kelch, Tub, PP2, Arm, and Cupin_8 subgroups contained 101, 107, 101, 41, 30, 15, 9, and 7 F-box genes, respectively, in the wheat genome. Fifteen F-box genes containing a small number of other specific domains (FBOs) were classified according to the functional domains, which included actin, LysM, Myb-binding, pro-isomerase, and WD40 (Figure 1 and Appendix A).

### 3.2. GO Analysis of F-Box Genes in the Wheat Genome

GO annotations were assigned using the Blast2GO bioinformatics platform to predict the functions of the wheat F-box genes in cellular metabolism. In total, 510 and 531 of the 1796 F-box genes were identified by mapping and annotation, respectively. The top three GO terms among the 1032 of the 1796 F-box genes were biological process, molecular functions, and cellular components (Appendix A). The results of the Fisher’s exact test (false discovery rate < 0.05, *p* < 0.05) showed the enrichment of several distinguishable GO terms in each of the three categories (Figure 2 and Appendix A). In total, 341 and 320 F-box genes were predicted under the terms SCF-dependent proteasomal ubiquitin-dependent protein catabolic process (GO:0031146) and protein ubiquitination (GO:0016567) of the biological process category, respectively, whereas 290 F-box genes were assigned to the integral component of the membrane (GO:0016021) in the cellular component category of the GO terms, followed by the nucleus (GO:0005634) and Cul3-RING ubiquitin ligase complex (GO:0031463). In the molecular function category, 363 F-box genes were predicted to be involved in ubiquitin–protein transferase activity (GO:0004842).

### 3.3. Phylogenetic Relationships and Chromosomal Localization of F-Box Genes

Phylogenetic analysis was conducted using the full-length protein sequences encoded by 1796 F-box genes (Appendix A). Because the FBX group containing only one FBX domain in the N-terminal was the most abundant group (1370 genes, 76% of F-box genes of wheat) and showed high sequence diversity, phylogenetic analysis of F-box proteins possessed one or more functional domains besides the FBX subgroup are shown in Figure 3. All F-box proteins containing DUF295, Cupin_8, Actin, and Arm domains were grouped together, whereas the F-box proteins possessing FBA, FBD, Kelch, and PP2 had scattered patterns within the F-box subfamilies and were grouped by F-box proteins with other domains. The FBO subfamilies with less than five members were clustered together in one subgroup. Using the IWGSC Reference Sequence v1.0 to determine chromosomal locations, there were 340, 323, and 299 F-box genes on chromosomes 3, 6, and 7, respectively. These chromosomes possessed relatively high numbers of F-box genes compared with the other chromosomes (Figure 4). Meanwhile, only 183 and 157 F-box genes were detected on chromosomes 1 and 4, respectively. Analysis of homoeologous chromosomes revealed that 517, 680, and 599 F-box genes were located in subgenomes A, B, and D, respectively. The distribution of F-box genes across chromosomes was not correlated with chromosome length (Pearson correlation, r = 0.285, *p* = 0.210).

### 3.4. Homology and Synteny Analysis of F-Box Genes in Wheat and Other Plants

To explore the genomic distribution of F-box genes based on sequence homology among homoeologous and non-homoeologous wheat chromosomes, the similarity between F-box genes and genes on other chromosomes was analyzed by comparing the protein sequences. Because of the high sequence homology between homoeologous chromosomes in wheat, the genomic distribution of F-box genes was identified using the BLASTP algorithm with strict parameter conditions, including an E-value of <1 × 10^−10^, minimum sequence identity of >90%, and bitscore of >500.

A high-resolution and comprehensive analysis of the wheat genome by the IWGSC revealed 88,733 high-confidence homologous genes of four major types (1:1:1, 1:1:0, 0:1:1, and 1:0:1) among the A, B, and D subgenomes of wheat [20]. Homoeologous F-box genes were analyzed with the information of IWGSC Reference Sequence v1.0 and were found to possess functional domains based on this study. A total of 921 types of homoeologous F-box genes were detected in the wheat genome. Among them, the 1:1:1, 1:1:0, 0:1:1, and 1:0:1 homoeologous groups of F-box genes were detected 332, 107, 179 and 120 times, respectively. The other minor types of homoeologous F-box genes, such as N:1:1, 1:N:1. 1:1:N, and N:0:1 were detected 213 times (Appendix A). In this study, 1311 wheat F-box gene pairs that were matched with the BLASTP algorithm with an E-value of <1 × 10^−10^, minimum sequence identity of >90%, and bitscore of >500 were used to create a synteny map (Figure 5A). Among them, 558 pairs of F-box genes were detected in two subgenomes, and 510 pairs of F-box genes were identified as 255 triplets in three loci of the wheat genome. Furthermore, 243 pairs of F-box genes had more than three loci (quadruplet) in the wheat genome (Figure 5A and Appendix A). Additionally, comparative synteny maps were constructed for wheat, *Brachypodium*, and rice to investigate the evolutionary history of the wheat F-box genes. Of the wheat F-box genes, 173 showed synteny with 67 *Brachypodium* F-box genes, comprising 17, 17, 15, 12, and 8 F-box genes on chromosomes 1, 2, 3, 4, and 5, respectively. F-box genes originated from most of the chromosomes in wheat, with the exception of subgenomes 1A and 6D, which were matched to 17 F-box genes on chromosome 1 of *Brachypodium*, whereas all the F-box genes on chromosome 5 of *Brachypodium* had high sequence similarity with the F-box genes on wheat chromosome 2 (Figure 5B and Appendix A). In the rice genome, 244 wheat F-box genes were matched to 81 syntenic rice genes. The maximum and minimum numbers of syntenic F-box genes in rice were 19 and 3 located in chromosomes 1 and 12, respectively. Interestingly, all 19 F-box genes on chromosome 1 of rice was matched to 36 F-box genes located on chromosome 3 of the wheat genome. Moreover, 4 and 3 F-box genes on chromosome 5 and 12 of rice showed synteny with 11 and 10 F-box genes on chromosome 5 of the wheat genome, respectively (Figure 5C and Appendix A).

### 3.5. Expression Profiling of Wheat F-Box Genes at Different Developmental Stages

RNA-Seq was conducted to reveal the expression profiles of F-box genes at seven different stages of the Zadoks growth scale in wheat. RPKM values were calculated for 808 F-box genes that are expressed during at least one developmental stage. Next, 209 of the genes with RPKM values of >1 in all developmental stages were divided into five groups based on their RPKM values for k-means clustering (Figure 6 and Appendix A). The 39 F-box genes in group 1 had lower expression levels in all development stages than the F-box genes in the other groups, whereas the 28 genes in group 5 were highly expressed in the wheat developmental stages. The highly expressed F-box genes in group 5 comprised 13 FBX, 7 Tub, 4 domain of unknown function (DUF), 2 Actin, and 2 Kelch subgroups. Among them, TraesCS2B01G570000.1:TraesCS2D01G541300.1 (Actin subgroup), TraesCS3A01G113600.1:TraesCS3D01G115900.1 (DUF subgroup), TraesCS1A01G033100.1: TraesCS1D01G034100.1 (FBX subgroup), TraesCS1A01G408700.1:TraesCS1D01G415800.1 (FBX subgroup), TraesCS5A01G075400.1:TraesCS5D01G088900.1 (FBX subgroup), TraesCS1A01G273100.1:TraesCS1B01G282900.1:TraesCS1D01G273200.1 (Tub subgroup), TraesCS2A01G574000.1:TraesCS2D01G584900.1 (Tub subgroup), and TraesCS4A01G209000.1:TraesCS4D01G110100.1 (Tub subgroup) were identified as homoeologous genes. The 33 F-box genes in group 2 were slightly highly expressed in spikelet developmental stages 4, 5, and 6. However, the expression levels of these genes decreased during ripening in Stage 7. Of the 33 F-box genes in group 2, 26, 2, 2, 1, 1, and 1 F-box genes were classified into the FBX, Kelch, FBD, Tub, Arm, and DUF subgroups, respectively. The 60 genes in group 3 were highly expressed in vegetative tissue, such as leaves at stages 1, 2, and 3, and spikelets at Stage 4; however, the expression levels decreased during spike development. Group 3 consisted of 25, 12, 9, 5, 3, 2, 1, 1, 1, and 1 in the FBX, Tub, Kelch, FBD, LysM, DUF295, Actin, Arm, FBA, and Myb-DNA-binding domain-containing subgroups, respectively. Conversely, the 49 genes in group 4 had relatively low expression levels in vegetative tissues and high expression levels during spike development. Of the 49 F-box genes in group 4, 34, 5, 3, 2, 1, 1 and 1 F-box genes were grouped by the FBX, FBD, Kelch, WD40, PP2, FBA, and Arm subgroups, respectively. In support of these results, 9 F-box genes in groups 2, 3, and 4 were selected for qRT-PCR (Figure 7). The results of qRT-PCR were consistent with those of RNA-Seq.

## 4. Discussion

The UPS is an important protein degradation mechanism involved in various cellular processes. Approximately 680–858 and 660–698 F-box genes have been reported in rice and *Arabidopsis*, respectively [35,36], although the numbers of F-box genes identified within species differ depending on the analytical method. F-box proteins play regulatory roles in protein degradation through the proteolytic mechanism of multi-protein E3 ubiquitin ligase in response to cellular signals during plant development and growth, hormone responses, and biotic/abiotic stress responses [2]. For example, in wheat, the TaFBA1 F-box protein with a FBA domain has been implicated in drought tolerance [37], whereas the F-box protein TaJAZ1 regulates resistance against powdery mildew [38], and the cyclin F-box domain TaCFBD gene is involved in inflorescence development and cold stress responses [39]. To date, however, a functional analysis of wheat F-box genes was conducted at the single gene level despite the important roles in several aspects of plant abiotic/biotic stress and development.

A genome-based study of the F-box genes in wheat was not previously possible because of the chromosome complexity and huge size of the hexaploid wheat genome. However, the establishment of a wheat reference genome sequence has recently changed this. Wheat has a large three-part genome (AA, BB, and DD, 2n = 6× = 42, approximately 17 Gbp). Therefore, it was predicted that the wheat genome would contain more F-box genes than the rice and *Arabidopsis* genomes. Here, 1796 F-box genes were identified in the wheat genome by HMMER analysis using the Pfam and InterPro databases, and a large number of these wheat F-box proteins (1370, 76%) contained only one functional domain (F-box or F-box-like domain). After FBX, the second most abundant domain was the FBD domain (107 genes), which plays a role in protein binding in *Arabidopsis* and is particularly highly expressed in vascular tissues via the auxin-independent pathway [40]. The 101 F-box proteins with FBA domains identified in the wheat genome mainly function in hormonal signal transduction as well as glycoprotein degradation in tomatoes [41] and drought tolerance in wheat [42]. All 101 proteins in the DUF subfamily contain a DUF295 domain, which is found in various plant species. One of the F-box proteins containing a DUF295 domain plays a role in the post-translational regulation of the curly leaf phenotype during post-germination growth in *Arabidopsis* [43,44]. Additionally, 41 F-box proteins with Kelch repeats, which are found in most organisms, were identified in the wheat genome. These proteins have regulatory roles in the circadian rhythm, flowering time, and seed germination of *Arabidopsis* [45,46,47]. Thirty F-box proteins with a Tub domain in the C-terminus were identified in the wheat genome in the present study. This domain may contain a binding motif for Cullin–RING ubiquitin ligases in rice and *Arabidopsis* [48]. Fifteen F-box proteins containing a PP2 domain were also identified in wheat. These proteins play a potential role in regulating transcript accumulation in seed development, although the function of the PP2 domain remains unknown in several other plant species.

Of the 1796 F-box genes, 1032 were classified into the top three GO terms—biological process, molecular functions, and cellular components. Among them, the terms SCF-dependent proteasomal ubiquitin-dependent protein catabolic process (341 F-box genes) and protein ubiquitination (320 F-box genes) in the biological process category and the term ubiquitin-like protein transferase activity in the molecular function category were over-represented according to GO analysis with Fisher’s exact test. This result indicates that F-box proteins play a regulatory role in protein degradation through the proteolytic mechanism of multi-protein E3 ubiquitin ligase.

F-box genes were found to be widespread throughout the whole wheat genome, and localization was not correlated with chromosome length. F-box genes have been predicted to exist in large numbers to regulate protein degradation, which is necessary for the regulation of various biological processes. BLASTP analysis of 1311 matched pairs of F-box genes (E-value < 1 × 10^−10^, minimum sequence identity > 90%, and bitscore > 500) was used to create a synteny map, which included 510 pairs, 255 triplets, and more than 243 quadruplet matches. Furthermore, homoeologous F-box genes in the wheat genome were identified based on the IWGSC Reference Sequence (v1.0). Synteny analysis with other plants identified 173 and 244 F-box genes in wheat that were closely related to the F-box genes of *B. distachyon* and rice, respectively. The phylogenetic position of *B. distachyon* indicates that it is suitable for use as a representative grass species with a large genome, such as the cool-season grasses wheat and rye. Rice is also an ideal grass model for research on numerous communities [21]. Orthologs of *B. distachyon* genes and F-box genes can be used for molecular and functional analyses to reveal the roles of F-box genes in crucial agronomic traits in winter crops.

A transcriptome analysis of wheat at various developmental stages was conducted to investigate the expression profiles of putative F-box genes. Transcript abundance was analyzed based on RPKM values, which revealed that some F-box genes were differentially expressed in one or more of the developmental stages of wheat. The expression patterns of F-box genes in groups 2, 3, and 4 (Figure 6) were specific to particular developmental stages. The expression patterns of TraesCS3A01G377800.1, TraesCS5A01G534800.1, and TraesCS7A01G559600.1 in group 2; TraesCS1D01G268400.1, TraesCS4A01G275700.1, and TraesCS6B01G275600.1 in group 3; TraesCS1B01G407200.1, TraesCS3D01G379700.1, and TraesCS6D01G130000.1 in group 4 were validated by qRT-PCR, and the results corroborated the RNA-Seq results, which confirmed developmental stage-specific expression in vegetative stages. Because the cellular and biological statuses greatly change during seed maturation and the grain filling stage, post-translational regulation is necessary to promptly activate phage conversion for seed development. In the present study, 49 F-box genes, including TraesCS1B01G407200.1, TraesCS3D01G379700.1, and TraesCS6D01G130000.1 in group 4, were found to be differentially expressed in at least one grain developmental stage in wheat, suggesting that these F-box genes are involved in protein regulation during seed maturation. By contrast, 60 F-box genes in group 3, including TraesCS1D01G268400.1, TraesCS4A01G275700.1, and TraesCS6B01G275600.1, were highly expressed in the vegetative stages rather than the seed development stages. Some F-box genes in group 3 with tissue-specific expressions participate in important functions in target tissues during plant growth in vegetative stages, similar to the corresponding F-box genes in rice and *Arabidopsis* [49,50]. Conversely, the continuously expressed F-box genes in groups 1 and 5 might be involved in general cellular processes, such as the regulation of protein homeostasis during plant development.

Here, a genome-based examination was conducted in a staple crop—wheat—and 1796 F-box genes were identified and classified according to specific domains. Because of the homology of the hexaploid wheat genome, most of the F-box genes had high sequence homology with F-box genes located on homoeologous chromosomes. Additionally, GO analysis revealed that F-box genes play roles in the regulation of various other genes, whereas the synteny analysis of rice and *B. distachyon* revealed the evolutionary history of the F-box genes. F-box genes were clustered according to the expression patterns in different developmental stages, which revealed that some F-box genes were highly expressed in the vegetative and/or seed development stages. Overall, the results of the present study provide fundamental information on F-box genes, including the sequences, domains, chromosomal locations, and synteny with other model plants, which will be valuable for further molecular studies of candidate F-box genes. Nonetheless, further functional studies on wheat F-box genes under biotic/abiotic stress conditions are necessary to understand the roles of these genes in the regulation of the proteasome pathway to improve biotic/abiotic stress resistance. Furthermore, functional analysis of the F-box proteins under a crucial developmental process, such as the regulation of vernalization, photoperiod, and flowering time, could be useful to elucidate the mechanisms underlying the essential agronomic traits of wheat.

## Figures and Tables

**Figure 1 genes-11-01154-f001:**
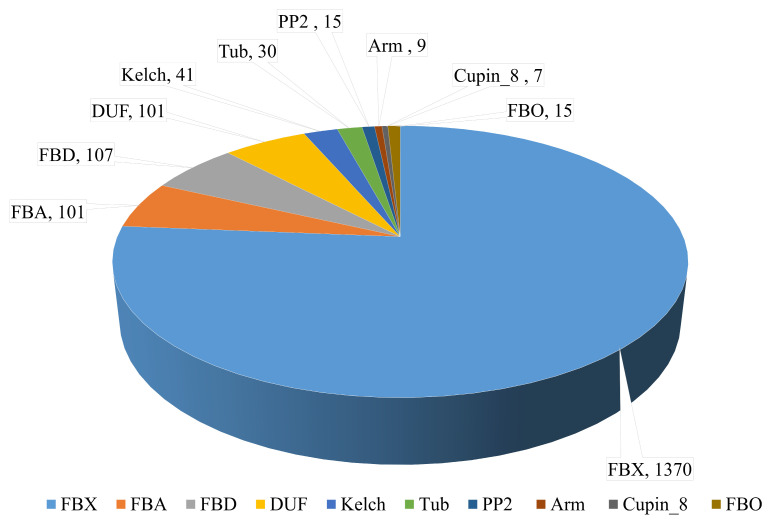
Classification of wheat F-box genes according to the functional domain. F-box (FBX): only FBX domain containing F-box genes. F-box-associated domain (FBA), FBD, DUF295, Kelch, Tubby (Tub), PP2, Arm, Cupin_8, and FBO (actin, LysM, Myb-binding, pro-isomerase, and WD40): FBX domains and other domains present in the peptide sequence. The number and type of F-box genes are shown.

**Figure 2 genes-11-01154-f002:**
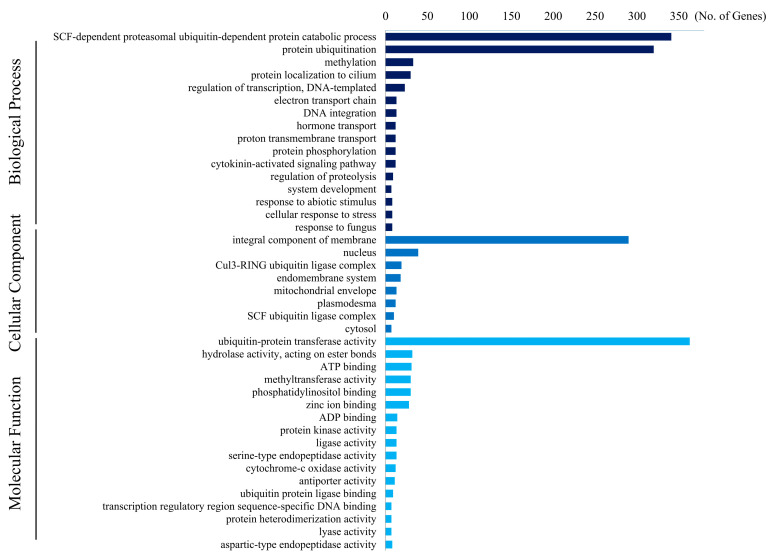
Enriched gene ontology terms of wheat F-box genes. Gene ontology (GO) analysis showed that the F-box genes were enriched in three categories (biology process, cellular component, and molecular function). Enrichment of GO terms was determined using Fisher’s exact test (false discovery rate < 0.05, *p* < 0.05).

**Figure 3 genes-11-01154-f003:**
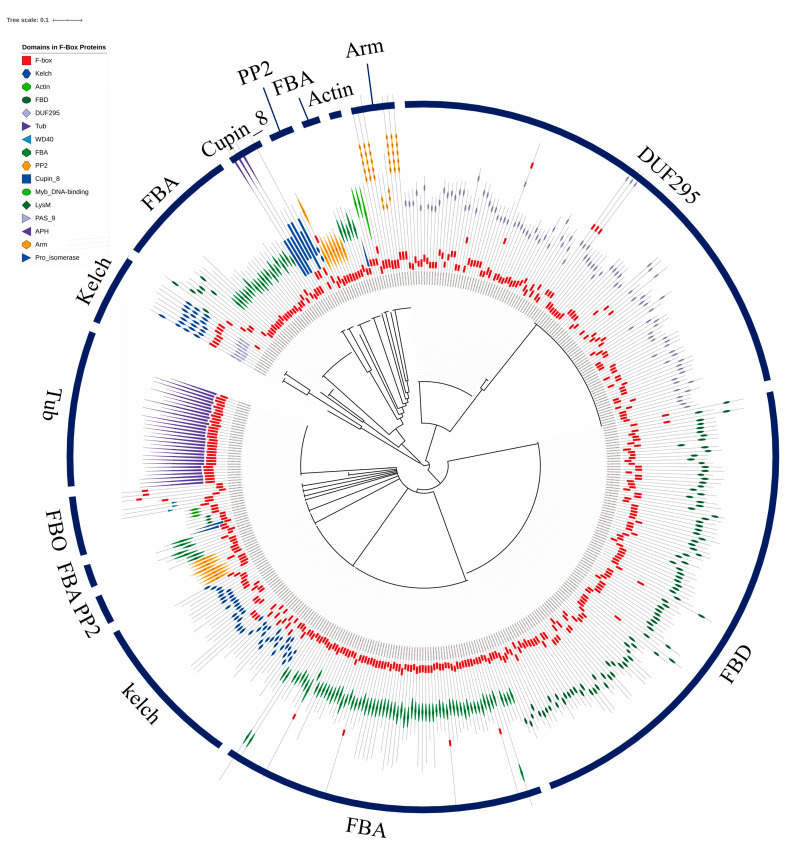
Phylogenic analysis of wheat F-box genes. A phylogenetic tree was generated using the Interactive Tree of Life (iTOL) tool with the amino acid sequences of F-box proteins that contained FBX domains in the N-terminal and one or more known functional domains in the C-terminal of the peptide sequences.

**Figure 4 genes-11-01154-f004:**
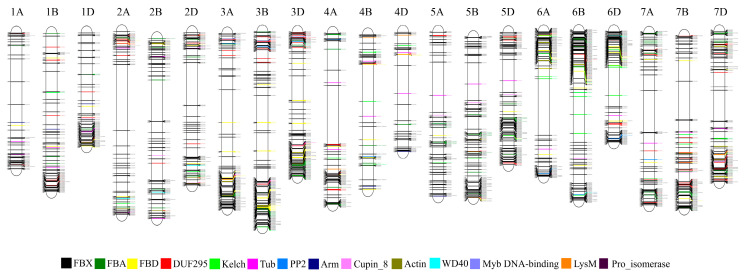
Chromosomal locations of wheat F-box genes. The positions of F-box genes (unit: Mb) are shown on the left of the chromosome, while the numbers on the right indicate the gene identifiers.

**Figure 5 genes-11-01154-f005:**
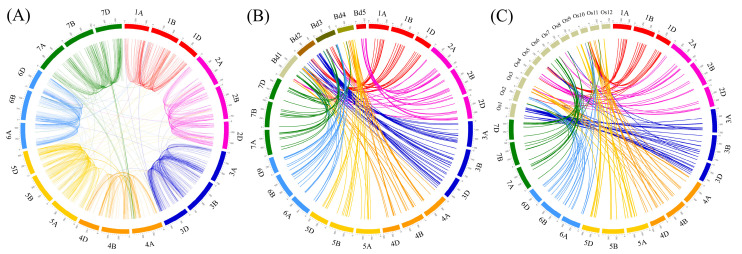
Synteny analysis of wheat F-box genes. (**A**) Synteny between F-box genes on the wheat homoeologous chromosome. F-box genes matched with the BLASTP algorithm with an E-value of <1 × 10^−10^, minimum sequence identity of >90%, and bitscore of >500 were used to create a synteny map of wheat F-box genes. Synteny between wheat (**B**) and *Brachypodium distachyon* and (**C**) rice. For synteny analysis of wheat against *Brachypodium* and rice, BLASTP was conducted with an E-value threshold of 1 × 10^−10^ and sequence identity of >80%. Colored lines denote syntenic regions between chickpea chromosomes and others. The genomes size of *Brachypodium* and rice were enlarged by a factor of 10.

**Figure 6 genes-11-01154-f006:**
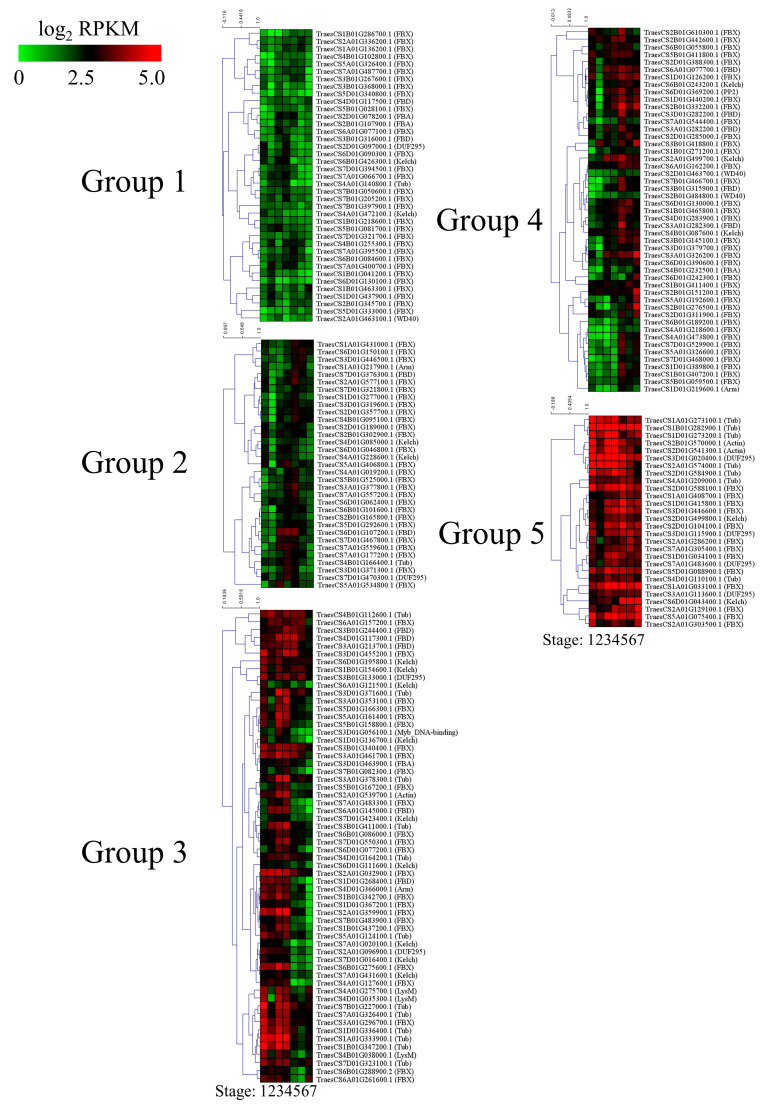
Expression profiling and classification of wheat F-box genes. K-means clustering of average log_2_ transformed reads per kilobase of transcript per million mapped reads (RPKM) values at different wheat developmental stages. Stage 1 (leaf; three leaves emerged, Z13), Stage 2 (leaf; main stem and four tillers present, Z24), Stage 3 (leaf; tip of ear just visible, booting stage, Z51), Stage 4 (spikelet; beginning of anthesis stage, Z61), Stage 5 (spikelet; early milk development stage, Z73), Stage 6 (spikelet; early dough stage, Z83), and Stage 7 (spikelet; hard grain stage, Z91) are shown. Z, Zadoks growth scale.

**Figure 7 genes-11-01154-f007:**
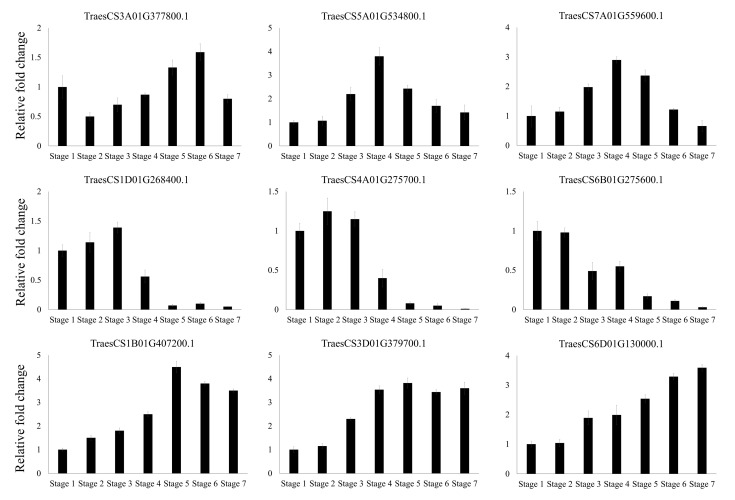
Validation of RNA-Seq results by qRT-PCR in wheat at different developmental stages. Stage 1 (leaf; three leaves emerged, Z13), Stage 2 (leaf; main stem and four tillers present, Z24), Stage 3 (leaf; tip of ear just visible, booting stage, Z51), Stage 4 (spikelet; beginning of anthesis stage, Z61), Stage 5 (spikelet; early milk development stage, Z73), Stage 6 (spikelet; early dough stage, Z83), and Stage 7 (spikelet; hard grain stage, Z91) are shown. Z, Zadoks growth scale. All samples were normalized to the *actin* gene. The data are presented as the average ± standard error (*n* = 3).

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
