# Peer review of "F-Box Genes in the Wheat Genome and Expression Profiling in Wheat at Different Developmental Stages"

_genes, 2020, doi:10.3390/genes11101154_

Round 1
Reviewer 1 Report
In general, this paper presents interesting information about the F-box gene family in Wheat. However, the manuscript lacks detail in several areas, and fails to inform the reader of why this gene family may be important in this species.
The quality of the English is generally good, however there are many typos, in particular a recurring mistake where spaces are absent between words.
The quality of the figures is poor, and the font size is so small that I couldn’t read it.
In terms of the materials and methods, I found there to be too little detail, especially in the RNA sequencing. The methods need to be more transparent, especially regarding the number of replicates and what quality checks were carried out. This issue must be addressed as it is a serious concern.
Abstract and Introduction:
Specific comments:
L13 – rhythm should be rhythms
L16 space between Triticum aestivum
L17 space missing
L21 Brachypodium should be written in full and italics
L33 revise sentence, not clear
L34 typos spaces missing
L37 enzyme singular
L40 space missing
L48 revise English
L57 ref for genome size
L57-61 revise this to more correctly introduce the wheat genome references
Introduction could be improved to better introduce F-box proteins and their role. Why are they interesting in wheat.
Materials and Methods
L73 reference for IWGSC
L78 were any parameters defined for HMMER? If so please state.
L78 by redundant do you mean duplicates? Were they found multiple times or present multiple times in the reference?
L84 phylogenetic analysis is very basic, were any substitution models used and where any steps taken to check the quality of the alignment? I would like to see evidence that the best quality phylogenetic tree was created as simply using ClustalW does not create the best tree. I suggest you read ‘A new phylogenetic protocol: dealing with model misspecification and confirmation bias in molecular phylogenetics’ by Jermiin et al https://doi.org/10.1093/nargab/lqaa041 for more information.
L96 why the change of thresholds between species?
L98 what does direction of the line mean?
L101 please introduce the genotypes in this section
Section 2.4 at Line 101 needs a lot more detail. How many plants of each genotype, how many reps etc? How many independent biological reps of each plant were used for RNA sequencing.
L113 were any steps taken to remove gDNA from RNA samples?
L114 was any quality check of RNA conducted (bioanalyzer, gel etc)?
L116 how were the raw RNAseq reads analysed and quality checked and then cleaned? Was any trimming done for adapters and low-quality sequencing? This section needs a lot more detail.
L128 how were the primers designed? Were they designed to be homoeologue specific or did they target all 3 copies of each gene? Much more detail needed here.
Results
L135 I’m confused here – you chose F-box genes with an FBX domain, and then go on to describe other subgroups of F-box genes. Please clarify this section in terms of the subgroupings you have used. Do all these genes have an FBX and other domains?
Section 3.2 please give more info about GO analysis – mapping/annotation etc
L149 what do you mean enriched? Do you mean that many of the F-box genes were associated with cellular process and metabolic process? I’m not sure that equates to enriched.
Did you carry out the enrichment test? If so, you need to state this before discussing enrichment or overrepresentation.
Figure 2 legend is lacking in a lot of detail
Line 160 section 3.3 this section on phylogenetic analysis is far too vague, it needs more detail on the subgroups. Were the F-box genes present as homoeologous groups, are there paralogues too? The figure 3 is not good enough quality to show anything interesting and the legend does not have enough detail. I suggest showing the overall tree, and also showing the phylogenetic trees of the subgroups, so that we can see the gene family in more detail.
Figure 4 is bad quality. It is impossible to read the gene names and the scale makes it such that its hard to see the chromosomes themselves. May I suggest that this figure is remade in mapchart using a common scale (i.e. one for the whole diagram rather than one on each chromosome), and the size of the chromosome is increased. Also, if the gene names were removed, it would be possible to see the location of the F-box genes. They couldn’t also be colour coded to match the colours used in the phylogenetic tree.
Also if homoeologous groups were put together, i.e. 1A, 1B and 1D in the top row, 2A, 2B, 2D in the second row etc, it would be a lot easier to see.
Section 3.4 Wheat homoeologues are already annotated and available on plant ensembl. Did you check if your inferred homoeologues matched what was previously annotated?
L191-193 I don’t understand this sentence please clarify it. Did you identify homoeologous pairs and triplets? Or just pairs. It would be interesting too see how many F-box genes are present as pairs and triplets as this is interesting from an evolutionary point of view.
L914-197 this is really interesting but lacks detail. I can see from the circus plots that large regions are syntenic, but this is not clear in the text.
Figure 5 is again poor quality. The text is impossible to read, and the legend lacks detail.
Section 3.5 the information is nice but the English could be improved.
Section 3.5 is there any relationship between the 5 groups and the type of F-box proteins they are? I.e. is there a relationship between the phylogenetic tree and the expression pattern?
Figure 6 is nice, but again the text is far too small. I suggest re-s-tacking the groups or rotating 90 degrees so that the text can be expanded.
The results in Figure 6 are very interesting – I would expect to see more detail about this, especially the high expression of group 5. What are these genes?
Figure 7 text is too small. No axis labels are present. Why are error bars SD rather than SE?
Discussion
Line 247 typo
L250 type
L272 correct reference
L294 what do you mean by differentially expressed? There is no differential expression analysis previously mentioned. What comparisons were made?
The discussion is generally good, but it also lacks context about where this study fits into the literature, what we already know about F-box genes in wheat, and why and how this study might be useful from an agronomic point of view.
Author Response
Abstract and Introduction
Reviewer comment:
- L13 – rhythm should be rhythms
- L16 space between Triticum aestivum
- L17 space missing
- L21 Brachypodium should be written in full and italics
- L33 revise sentence, not clear
- L34 typos spaces missing
- L37 enzyme singular
- L40 space missing
- L48 revise English
- L57 ref for genome size
Response:
- The typographical or spacing issues have been corrected throughout the manuscript.
Reviewer comment:
- L57-61 revise this to more correctly introduce the wheat genome references:
Response:
- The introduction of wheat genome has been revised with correct references as follows.
- Wheat is allohexaploid (2n = 6x = 42; AABBDD) with a huge genome of approximately 17 Gbp that consists of three closely related homoeologous subgenomes comprising a high percentage of repetitive sequences and homoeologous DNA copies. A draft genome of wheat referred to as the International Wheat Genome Sequencing Consortium (IWGSC) chromosome survey sequence assembly (IWGSC 2014) was provided to the researcher [18], followed by the wheat assembly TGACv1, which provides chromosome arm-specific chromosome survey sequence reads [19]. (Line 66-71)
Reviewer comment:
- Introduction could be improved to better introduce F-box proteins and their role. Why are they interesting in wheat.
Response:
- The following two paragraphs that contain the explanation of F-box proteins and their roles have been includedin the INTRODUCTION section of the revised manuscript.
- The SCF complex, which comprises the Skp1, Cullin, Rbx1, and F-box proteins, has been characterized in several species [7]. Cullin, the major structural scaffold for the SCF complex, connects Skp1 to Rbx1, and Skp1 binds F-box proteins via a substrate recognition function [8]. Numerous F-box proteins contain a conserved F-box (FBX) domain located in the 60 amino acids of the N-terminus. The F-box proteins can be classified by different C-terminal protein–protein interaction domains that recruit targets for proteasomal degradation. The C-terminal region of F-box proteins contains several protein–protein interaction domains, such as leucine-rich repeats, WD40, Kelch repeats, F-box-associated domain (FBA), or Tubby (Tub), which binds with target substrates and confers specificity to the F-box proteins [9].(Line 47-55)
- Various members of the F-box gene family have been identified in the genomes of different plant species. At least 678, 913, 480, and 656 F-box proteins have been reported in Oryza sativa, Medicago truncatula, Glycine max, and Arabidopsis thaliana, respectively [10]. The presence of high numbers of F-box proteins in plant genomes suggests that numerous SCF complexes regulate various substrates and play specific regulatory roles through post-translational modification in several biological processes, such as the regulation of flowering and circadian rhythms [11], photomorphogenesis [12], leaf senescence [13], floral development [14], lateral shoot branching [15], gibberellic acid signaling [16], and auxin signaling [17]. Functional studies of F-box proteins in Arabidopsis are ongoing; however, little is known about the F-box proteins of wheat. In addition, the F-box genes of wheat have not been classified according to the C-terminal domain structures. (Line 56-65)
Materials and Methods
Reviewer comment:
- L73 reference for IWGSC:
Response:
- The reference to IWGSC has been added in the revised manuscript. (Line 85-86)
Reviewer comment:
- L78 were any parameters defined for HMMER? If so please state.
Response:
- HMMER3 was used for profiling of the F-box proteins in the seed files with default parameters. The explanation of HMMER3 has been revised as follows.
- HMM profiling of the F-box proteins was conducted with the HMM files of F-box (PF00646), F-box–like (PF12937), F-box–like2 (PF13013), FBA1 (PF07734), FBA2 (PF07735), FBA3 (PF08268), and FBD (PF08387) domains, which were provided by Pfam [22] and searched against a protein databases of the wheat genome using the HMMER3 tool with default parameters [23]. (Line 87-91)
Reviewer comment:
- L78 by redundant do you mean duplicates? Were they found multiple times or present multiple times in the reference?
Response:
- Most of the F-box genes contained an F-box domain at the C-terminal, although some genes contained other domains, such as FBA and FBD. The ambiguous sentence has been revised as follows.
- Redundant F-box genes that did not contain an F-box domain in the C-terminal were removed, whereas F-box genes with a C-terminal FBX were verified using the SMART tool [24] and the Pfam protein database with an E-value threshold of 1e−5 for significance. (Line 91-94)
Reviewer comment:
- L84 phylogenetic analysis is very basic, were any substitution models used and where any steps taken to check the quality of the alignment? I would like to see evidence that the best quality phylogenetic tree was created as simply using ClustalW does not create the best tree. I suggest you read ‘A new phylogenetic protocol: dealing with model misspecification and confirmation bias in molecular phylogenetics’ by Jermiin et al https://doi.org/10.1093/nargab/lqaa041 for more information.
Response:
- Thank you for your advice. The phylogenetic tree was reconstructed with contacting functional domains of the Interactive Tree of Life (iTOL) and had been indicated by Figure 3 in the revised manuscript.
Reviewer comment:
- L96 why the change of thresholds between species?
Response:
- The homoeologous DNA copies from the three subgenomes (AA, BB, and DD) showed high sequence similarity; therefore, we used and adapted strict sequence similarity of the protein (>90%). For other plant species (Brachypodium and rice), > 80% peptide sequence similarity were used for synteny analysis. The followed paragraph has been edited in the revised manuscript.
- The BLASTP algorithm was used to predict the sequence homology of F-box genes to genes on homoeologous chromosomes in wheat, with an E-value threshold of<1e−10, sequence identity of >90%, and bitscore of >500, as the homoeologous DNA copies from the three subgenomes (AA, BB, and DD) had high sequence similarity. (Line 112-116)
Reviewer comment
- L98 what does direction of the line mean?
Response:
- The ambiguoussentence has been deleted from the revised manuscript (Line 116-118)
Reviewer comment
- L101 please introduce the genotypes in this section?
Response:
- One genotype of the developed RILs was selected. Therefore, “RILs” was revised to “common wheat” in the revised manuscript. An explanation of the type of wheat used in this study has been added to the revised manuscript as follows:
- Common wheat used in the present study was developed from a single plant descended from the F10 generation of a cross of Woori‐mil (Korea RDA accession no. IT172221) × D‐7 (an inbred line developed by Korea University, Fleming*4 /3/ PIO 2580 // T83103 *2 / Hamlet) as previously reported [29]. (Line 120-123)
Reviewer comment:
- Section 2.4 at Line 101 needs a lot more detail. How many plants of each genotype, how many reps etc? How many independent biological reps of each plant were used for RNA sequencing.
Response:
- We used one genotype of developed RILs and collected samples of three independent biological replicates in each developmental stage. Section 2.4 has been revised as follows.
- Seeds were vernalized at 4°C for 4 weeks to synchronize growth and then transferred to an Incu Tissue plant culture vessel (72 × 72 × 22 mm3; SPL Life Sciences, Gyeonggi-do, Korea) containing a polypropylene net floating on Hoagland solution (Sigma-Aldrich, St. Louis, USA). The plants were grown under controlled environments at 23°C–26°C and a 16-h light/8-h dark cycle. Using the Zadoks growth scale (Z) [30] as a reference, three independent biological replicatesof the following plant samples were collected at different developmental stages: leaves at Z13 (three leaves emerged; Stage 1), leaves at Z24 (main stem and four tillers present; Stage 2), leaves at Z51 (leaf at the tip of ear just visible, booting stage; Stage 3), spikelets at Z61 (beginning of anthesis; Stage 4), spikelets at Z73 (early milk development; Stage 5), spikelets at Z83 (early dough stage; Stage 6), and spikelets at Z91 (hard grain stage; Stage 7). Each collected sample was stored at −80°C until analysis. (Line 123-133)
Reviewer comment:
L113 were any steps taken to remove gDNA from RNA samples?
L114 was any quality check of RNA conducted (bioanalyzer, gel etc)?: L116 how were the raw RNAseq reads analysed and quality checked and then cleaned? Was any trimming done for adapters and low-quality sequencing? This section needs a lot more detail.
Response:
- The RNA sequencing method insection2.5 has been revised as follows:
- Total RNA was extracted from three independent biological replicates at each developmental stage using TRIzol reagent (Invitrogen, Waltham, MA, USA) and treated with DNase I to eliminate any contaminating genomic DNA. RNA quality was assessed using the Agilent 2100 bioanalyzer (Agilent Technologies, Amstelveen, The Netherlands), and RNA quantification was performed using ND-2000 Spectrophotometer (Thermo Inc., DE, USA). Total RNA (10 μg) extracted from the samples was used to construct RNA sequencing (RNA-Seq) paired-end libraries with the TruSeq RNA Sample Preparation Kit (catalog #RS-122-2001; Illumina, CA, USA). mRNA was isolated using the Poly(A) RNA Selection Kit (LEXOGEN, Inc., Austria) and reverse transcribed into cDNA in accordance with the manufacturer’s instructions. The Agilent 2100 bioanalyzer was used to check the libraries, and the DNA High Sensitivity Kit was used to evaluate the mean fragment size. High-throughput sequencing was performed using the HiSeq 2000 platform (Illumina, Inc., USA). The adaptor sequences were removed and sequence quality was tested using the BBduk tool (minimum length>20, and Q>20) before alignment. (Line 135-147)
Reviewer comment:
- L128 how were the primers designed? Were they designed to be homoeologue specific or did they target all 3 copies of each gene? Much more detail needed here.
Response:
- Primers were designed based on k-means clustering of the RNA-seq results. For primer design, the sequences of the F-box proteins selected by k-means clustering methods were blasted against the wheat reference genome to compare homoeologous gene sequences. The homoeologous specific region of the selected FBX genes was used to design primers for the validation by qRT-PCR.
- The sequences of the F-box proteins selected by k-means clustering methods were blasted against the wheat reference genome to compare genes with homoeologous sequences. Homoeologous regions specific to selected F-box genes were used to design gene-specific primers for the validation of the qRT-PCR products (Table S1). (Line 158-159)
Results
Reviewer comment:
- L135 I’m confused here – you chose F-box genes with an FBX domain, and then go on to describe other subgroups of F-box genes. Please clarify this section in terms of the subgroupings you have used. Do all these genes have an FBX and other domains?
Response:
- A total of 1,796 F-box genes containing an FBX domain were selected because 1,370 F-box genes did not contain other functional domains, such as FBA, FBD, DUF295, Kelch, Tub, PP2, Arm, and Cupin_8 domains but an FBX domain in the C-terminal. We referred to F-box genes that contained only an FBX domain.
The followed sentence was inserted in the revised manuscript.
A total of 1,796 F-box genes containing an FBX domain were selected for further study (Table S2). The F-box genes were classified into 10 subgroups containing an FBX domain and other specific functional domains. Genes containing an FBX domain in the N-terminal of the peptide sequence were the most abundant gene subgroup of the wheat genome (1,370 genes). The groups of other F-box genes are referred to as FBA, FBD, DUF295, Kelch, Tub, PP2, Arm, and Cupin_8. Each of these subgroups contained FBX domains in the N-terminal and one or more known functional domains in the C-terminal of the peptide sequences. The FBA, FBD, DUF295, Kelch, Tub, PP2, Arm, and Cupin_8 subgroups contained 101, 107, 101, 41, 30, 15, 9, and 7 F-box genes, respectively, in the wheat genome. (Line 168-177)
Reviewer comment:
- Section 3.2 please give more info about GO analysis – mapping/annotation etc
Response:
- GO analysis was conducted with the Blast2GO program. Local BlastX was conducted with the peptide sequences of the Poaceae family retrieved from the NCBI database to initiate GO analysis for construction of XML files. Next, the mapping step were executed to retrieve the gene names, GO terms, and protein sequences from the UniProt database, which was followed by annotation with default parameters (e-value< 1.0e-6, annotation cutoff = 55, GO weight = 5). The followed text was added to the Materials and Methods and Results sections. In addition, the result of Blast2GO has been added in Supplementary Table S3 of the revised manuscript.
- Local BlastX was conducted with peptide sequences of the Poaceae family sequences retrieved from the National Center for Biotechnology Information database to initiate gene ontology (GO) analysis for the construction of xml files. Next, mapping was performed to retrieve the gene names, GO terms, and protein sequences from the UniProt database, which was followed by annotation with default parameters (e-value<1.0e-6, annotation cutoff =55, GO weight=5). (Line 95-100)
- GO annotations were assigned using the Blast2GO bioinformatics platform to predict the functions of the wheat F-box genes in cellular metabolism. In total, 510 and 531 of the 1,796 F-box genes were identified by mapping and annotation, respectively. (Line 186-188)
Reviewer comment:
- L149 what do you mean enriched? Do you mean that many of the F-box genes were associated with cellular process and metabolic process? I’m not sure that equates to enriched.
- Did you carry out the enrichment test? If so, you need to state this before discussing enrichment or overrepresentation.
Response:
- In accordance with the Reviewer’s comments, the result of enrichment analysis of GO terms, as determined with the Fisher’s exact test, are presented in Figure 2 of the revised manuscript. The followingtext was added to section 3.2:
- The top three GO terms among the 1,032 of the 1,796 F-box genes were biological process, molecular functions, and cellular components (Table S3). The results of the Fisher’s exact test (false discovery rate<0.05, p<0.05) showed the enrichment of several distinguishable GO terms in each of the three categories (Figure 2 and TableS4). In total, 341 and 320 F-box genes were predicted under the terms SCF-dependent proteasomal ubiquitin-dependent protein catabolic process (GO:0031146) and protein ubiquitination (GO:0016567) of the biological process category, respectively, whereas 290 F-box genes were assigned to integral component of membrane (GO:0016021) in the cellular component category of the GO terms, followed by nucleus (GO:0005634) and Cul3-RING ubiquitin ligase complex (GO:0031463). In the molecular function category, 363 F-box genes were predicted to be involved in ubiquitin-protein transferase activity (GO:0004842). (Line 188-198)
Reviewer comment:
- Figure 2 legend is lacking in a lot of detail
Response:
- The legend of Figure 2 has been revised as follows:
- Figure 2. Enriched gene ontology terms of wheat F-box genes. GO analysis showed that the F-box genes were enriched in three categories (biology process, cellular component, and molecular function). Enrichment of GO terms was determined using the Fisher’s exact test (false discovery rate<0.05, p<0.05). (Line 200-202)
Reviewer comment:
- Line 160 section 3.3 this section on phylogenetic analysis is far too vague, it needs more detail on the subgroups. Were the F-box genes present as homoeologous groups, are there paralogues too? The figure 3 is not good enough quality to show anything interesting and the legend does not have enough detail. I suggest showing the overall tree, and also showing the phylogenetic trees of the subgroups, so that we can see the gene family in more detail.
Response:
- The domains of the F-box genes are presented in Figure 3 of the revised manuscript. The most abundant FBX group (1,370 genes) containing only the FBX domain in the N-terminal was excluded from the phylogenetic analysis. The text in section 3.3 and legend of Figure 3 have been revised as follows:
- Phylogenetic analysis was conducted using the full-length protein sequences encoded by 1,796 F-box genes (Supplementary Figure 1). Because the FBX group containing only one FBX domain in the N-terminal was the most abundant group (1,370 genes, 76% of F-box genes ofwheat) and showed high sequence diversity, phylogenetic analysis of F-box proteins possessed one or more functional domains besides FBX subgroup were showed in Figure 3. All F-box proteins containing DUF295, Cupin_8, Actin, and Arm domains were grouped together, whereas the F-box proteins possessing FBA, FBD, Kelch, and PP2 had scattered patterns within the F-box subfamilies and were grouped by proteins with other domains. The FBO subfamilies with less than five members were clustered together in one subgroup. (Line 204-212)Figure 3. Phylogenic analysis of wheat F-box genes. A phylogenetic tree was generated using the iTOL tool with the amino acid sequences of F-box proteins that contained FBX domains in the N-terminal and one or more known functional domains in the C-terminal of the peptide sequences. (Line 222-224)
- In addition, information on the homoeologous genes in the F-box family was added to TableS5 based on the IWGSC Reference Sequence v1.0, and a description was added to section 3.4
Reviewer comment:
- Figure 4 is bad quality. It is impossible to read the gene names and the scale makes it such that its hard to see the chromosomes themselves. May I suggest that this figure is remade in mapchart using a common scale (i.e. one for the whole diagram rather than one on each chromosome), and the size of the chromosome is increased. Also, if the gene names were removed, it would be possible to see the location of the F-box genes. They couldn’t also be colour coded to match the colours used in the phylogenetic tree.
- Also if homoeologous groups were put together, i.e. 1A, 1B and 1D in the top row, 2A, 2B, 2D in the second row etc, it would be a lot easier to see.
Response:
- In accordance with the reviewer’s comment, Figure 4 was reconstructed in the revised manuscript. The chromosome size was increased, and the size of gene name was reduced to better visualize the locations of the F-box genes. In addition, the chromosome locations of each subfamily of F-box proteins classified by the domain (e.g., FBX, FBA, FBD, and DUF) are indicated by different colors in Figure4. Because there were more than 1700 F-box genes, it was difficult to indicate the homoeologous genes in Figure 4. Therefore, information regarding the homoeologous F-box genes was added to TableS5 based on the IWGSC Reference Sequence v1.0 and descriptions were added to section 3.4
Reviewer comment:
- Section 3.4 Wheat homoeologues are already annotated and available on plant ensembl. Did you check if your inferred homoeologues matched what was previously annotated?
- L191-193 I don’t understand this sentence please clarify it. Did you identify homoeologous pairs and triplets? Or just pairs. It would be interesting too see how many F-box genes are present as pairs and triplets as this is interesting from an evolutionary point of view.
Response:
- We compared the 1,796 F-box genes identified in this study with the information on homoeologous pairs (IWGSC Reference Sequence v1.0). However, in many cases of IWGSC reference genome, when one F-box genes contained F-box domain, the homoeologous genes did not contain an F-box domain. Here, we present some examples of interprocsan results comparing the functional domains of F-box genes and homoeologous genes annotated in the IWSGSC reference genome.
- Example 1: TraesCS1A01G408700, TraesCS1B01G437400, and TraesCS1D01G415800 are annotated as homoeologous genes in each subgenome by IWGSC, but TraesCS1B01G437400 did not contain F-box domain (Reviewer comment Fig. 1).
Reviewer comment Fig. 1. TraesCS1A01G408700, TraesCS1B01G437400, and TraesCS1D01G415800 are annotated to homoeologous genes in each subgenome by IWGSC. Red boxes indicates F-box domain.
- Example 2: TraesCS2A01G049800, TraesCS2B01G554200, and TraesCS2D01G048900 are annotated as homoeologous genes in each subgenome by IWGSC, but TraesCS2B01G554200did not contain an F-box domain (Reviewer comment Fig. 2).
Reviewer comment Fig. 2. TraesCS2A01G049800, TraesCS2B01G554200, and TraesCS2D01G048900 are annotated to homoeologous genes in each subgenome by IWGSC. Red boxes indicates F-box domain.
- Example 3: TraesCS3A01G082000, TraesCS3B01G101100, and TraesCS3D01G065800 are annotated as homoeologous genes in each subgenome by IWGSC, but TraesCS3B01G101100and TraesCS3D01G065800 did not contain an F-box domain (Reviewer comment Fig. 3).
Reviewer comment Fig. 3. TraesCS3A01G082000, TraesCS3B01G101100, and TraesCS3D01G065800 are annotated to homoeologous genes in each subgenome by IWGSC. Red boxes indicates F-box domain.
- We collected and selected F-box proteins using the HMM profiling as the first selection, and then the F-box genes were verified with the use of SMART and Pfam. Therefore, 1,796 F-box genes were compared to homoeologous F-box genes of each subgenome from the IWGSC reference genome, which confirmed the homoeologous F-box genes identified in the present study. The following text, including the homoeologous F-box genes and synteny of F-box genes, was added to the revised manuscript:
- High-resolution and comprehensive analysis of the wheat genome by the IWGSC revealed a total of 88,733 high-confidence homologous genes of four major types (1:1:1, 1:1:0, 0:1:1, and 1:0:1) among the A, B, and D subgenomes of wheat [20]. Homoeologous F-box genes were analyzed with the information of IWGSC Reference Sequence v1.0 and possessing functional domains based on this study. A total of 921 types of homoeologous F-box genes were detected in the wheat genome. Among them, the 1:1:1, 1:1:0, 0:1:1, and 1:0:1 homoeologous groups of F-box genes were detected 332, 107, 179 and 120 times, respectively. The other minor types of homoeologous F-box genes, such as N:1:1, 1:N:1. 1:1:N, and N:0:1 were detected 213 times (Table S5). In this study, 1,311 wheat F-box gene pairs that were matched with the BLASTP algorithm with an E-value of <1e−10, minimum sequence identity of >90%, and bitscore of >500 were used to create a synteny map (Figure 5A). Among them, 558 pairs of F-box genes were detected in two subgenomes, and 510 pairs of F-box genes were identified as 255 triplets in three loci of the wheat genome. Furthermore, 243 pairs of F-box genes had more than three loci (quadruplet) in the wheat genome (Figure 5A and Table S6). (Line 236-248)
Reviewer comment:
- 914-197 this is really interesting but lacks detail. I can see from the circus plots that large regions are syntenic, but this is not clear in the text.
Response:
- A detailed explanation of the synteny analysis of wheat F-box genes with Brachypodium and rice was added to the Results section as follows:
- Additionally, comparative synteny maps were constructed for wheat, Brachypodium, and rice to investigate the evolutionary history of the wheat F-box genes. Of the wheat F-box genes, 173 showed synteny with 67 Brachypodium F-box genes, comprising 17, 17, 15, 12, and 8 F-box genes on chromosomes 1, 2, 3, 4, and 5, respectively. F-box genes originated from most of the chromosomes in wheat, with the exception of subgenomes 1A and 6D, which were matched to 17 F-box genes on chromosome 1 of Brachypodium, whereas all the F-box genes on chromosome 5 of Brachypodium had high sequence similarity with the F-box genes on wheat chromosome 2 (Figure 5B and Table S7). In the rice genome, 244 wheat F-box genes were matched to 81 syntenic rice genes. The maximum and minimum numbers of syntenic F-box genes in rice were 19 and 3 located in chromosomes 1 and 12, respectively. Interestingly, all 19 F-box genes in rice were on chromosome 1. The most abundant F-box gene of rice was matched to 36 F-box genes located on chromosome 3 of the wheat genome. Moreover, 4 and 3 F-box genes on chromosome 5 and 12 of rice showed synteny with 11 and 10 F-box genes on chromosome 5 of the wheat genome, respectively (Figure 5C and Table S8). (Line 249-261)
Reviewer comment:
- Figure 5 is again poor quality. The text is impossible to read, and the legend lacks detail.
Response:
- Figure 5 has been revised. In addition, the legend of Figure 5 has been revised as follows
- Figure 5. Synteny analysis of wheat F-box genes. (A) Synteny between F-box genes on the wheat homoeologous chromosome. F-box genes matched with the BLASTP algorithm with an E-value of <1e−10, minimum sequence identity of >90%, and bitscore of >500 were used to create a synteny map of wheat F-box genes. Synteny between wheat (B) and Brachypodium distachyon and (C) rice. For synteny analysis of wheat against Brachypodium and rice, BLASTP was conducted with an E-value threshold of 1e−10 and sequence identity of >80%.Colored lines denote syntenic regions between chickpea chromosomes and others. The genomes size of Brachypodium and rice were enlarged by a factor of 10.
Reviewer comment:
- Section 3.5 the information is nice but the English could be improved.
- Section 3.5 is there any relationship between the 5 groups and the type of F-box proteins they are? I.e. is there a relationship between the phylogenetic tree and the expression pattern?
- The results in Figure 6 are very interesting – I would expect to see more detail about this, especially the high expression of group 5. What are these genes?
Response:
- The paragraph in section 3.5 has been revised as follows:
- RNA-Seq was conducted to reveal the expression profiles of F-box genes at seven different stages of the Zadoks growth scale in wheat. RPKM values were calculated for 808 F-box genes that are expressed during at least one developmental stage. Next, 209 of the genes with RPKM values of >1 in all developmental stages were divided into five groups based on their RPKM values for k-means clustering (Figure 6 and Table S9). The 39 F-box genes in group 1 had lower expression levels in all development stages than the F-box genes in the other groups, whereas the 28 genes in group 5 were highly expressed in the wheat developmental stages. The highly expressed F-box genes in group 5 comprised 13 FBX, 7 Tub, 4 domain of unknown function (DUF), 2 Actin, and 2 Kelch subgroups. Among them, TraesCS2B01G570000.1:TraesCS2D01G541300.1 (Actin subgroup), TraesCS3A01G113600.1:TraesCS3D01G115900.1 (DUF subgroup), TraesCS1A01G033100.1: TraesCS1D01G034100.1 (FBX subgroup), TraesCS1A01G408700.1:TraesCS1D01G415800.1 (FBX subgroup), TraesCS5A01G075400.1:TraesCS5D01G088900.1 (FBX subgroup), TraesCS1A01G273100.1:TraesCS1B01G282900.1:TraesCS1D01G273200.1 (Tub subgroup), TraesCS2A01G574000.1:TraesCS2D01G584900.1 (Tub subgroup), and TraesCS4A01G209000.1:TraesCS4D01G110100.1 (Tub subgroup) were identified as homoeologous genes. The 33 F-box genes in group 2 were slightly highly expressed in spikelet developmental stages 4, 5, and 6. However, the expression levels of these genes decreased during ripening in stage 7. Of the 33 F-box genes in group 2, 26, 2, 2, 1, 1, and 1 were in the FBX, Kelch, FBD, Tub, Arm, and DUF subgroups, respectively. The 60 genes in group 3 were highly expressed in vegetative tissue, such as leaves at stages 1, 2, and 3, and spikelets at stage 4; however, the expression levels decreased during spike development. Group 3 consisted of 25, 12, 9, 5, 3, 2, 1, 1, 1, and 1 in the FBX, Tub, Kelch, FBD, LysM, DUF295, Actin, Arm, FBA, and Myb-DNA binding domain-containing subgroups, respectively. Conversely, the 49 genes in group 4 had relatively low expression levels in vegetative tissues and high expression levels during spike development. Of the 49 F-box genes in group 4, 34, 5, 3, 2, 1, 1 and 1 were in the FBX, FBD, Kelch, WD40, PP2, FBA, and Arm subgroups, respectively. In support of these results, 9 F-box genes in groups 2, 3, and 4 were selected for qRT-PCR (Figure 7). The results of qRT-PCR were consistent with those of RNA-Seq. (Line 273-299)
Reviewer comment:
- Figure 6 is nice, but again the text is far too small. I suggest re-s-tacking the groups or rotating 90 degrees so that the text can be expanded.
Response:
- In accordance with the reviewer’s comment, Figure 6 has been revised.
Reviewer comment:
- Figure 7 text is too small. No axis labels are present. Why are error bars SD rather than SE?
Response:
- Text and axis labels in Figure 7 have been revised accordingly. Figure7 has been revised, and SD has been changed to SE in the legend.
Discussion
Reviewer comment:
- Line 247 typo
- L250 type
- L272 correct reference
Response:
- The typographical errors and reference issues have been corrected in the revised manuscript.
Reviewer comment:
- L294 what do you mean by differentially expressed? There is no differential expression analysis previously mentioned. What comparisons were made?
Response:
- The term “expressed differentially” was revised to “showed high expression” in the revised manuscript.
Reviewer comment:
- The discussion is generally good, but it also lacks context about where this study fits into the literature, what we already know about F-box genes in wheat, and why and how this study might be useful from an agronomic point of view.
Response:
- The Discussion section has been revised as follows:
- F-box proteins play regulatory roles in protein degradation through the proteolytic mechanism of multi-protein E3 ubiquitin ligase in response to cellular signals during plant development and growth, hormone responses, and biotic/abiotic stress responses [2]. For example, in wheat, the TaFBA1 F-box protein with a FBA domain has been implicated in drought tolerance [36], whereas the F-box protein TaJAZ1 regulates resistance against powdery mildew [37], and the cyclin F-box domain TaCFBD gene is involved in inflorescence development and cold stress response [38]. To date, however, functional analysis of wheat F-box genes was conducted at the single gene level despite the important roles in several aspects of plant abiotic/biotic stress and development. (Line 318-326)
- Of the 1,796 F-box genes, 1,032 were classified into the top three GO terms biological process, molecular functions, and cellular components. Among them, the terms SCF-dependent proteasomal ubiquitin-dependent protein catabolic process (341 F-box genes) and protein ubiquitination (320 F-box genes) in the biological process category and the term ubiquitin-like protein transferase activity in the molecular function category were over represented according to GO analysis with the Fisher’s exact test. (Line 350-355)
- BLASTP analysis of 1,311 matched pairs of F-box genes (E-value<1e−10, minimum sequence identity>90%, and bitscore>500) was used to create a synteny map, which included 510 pairs, 255 triplets, and more than 243 quadruplet matches. Furthermore, homoeologous F-box genes in the wheat genome were identified based on the IWGSC Reference Sequence (v1.0). Synteny analysis with other plants identified 173 and 244 F-box genes in wheat that were closely related to the F-box genes of Brachypodium distachyon and rice, respectively. The phylogenetic position of Brachypodium distachyon indicates that it is suitable for use as a representative grass species with a large genome, such as the cool-season grasses wheat and rye. Rice is also an ideal grass model for research on numerous communities [21]. Orthologs of Brachypodium distachyon genes and F-box genes can be used for molecular and functional analyses to reveal the roles of F-box genes in crucial agronomic traits in winter crops. (Line 350-370)
- Nonetheless, further functional studies on wheat F-box genes under biotic/abiotic stress conditions are necessary to understand the roles of these genes in the regulation of the proteasome pathway to improve biotic/abiotic stress resistance. Furthermore, functio

Reviewer 2 Report
Authors did a good job in identifying and characterizing the Fox genes in wheat. But to make it different and novel from other studies of gene family characterization authors need to add more in results and discussion.
Comments:
- Please improve the English through out the manuscript. There are many places where words are jumbled around or have space issues.
- In the Abstract authors have mentioned about synteny and evolution but I have not seen enough related to evolution in the result and discussion.
- What is the status of gene duplication in F-box gene family?
- In the phylogeny, Is different groups have some characteristics for example with respect to domain presence and expression pattern? What can be possible explanation for their functions based on these results?
Author Response
Reviewer comment:
- Please improve the English throughout the manuscript. There are many places where words are jumbled around or have space issues.
Response:
- The typographical errors and spacing issues have been corrected throughout the manuscript.
Reviewer comment:
- In the Abstract authors have mentioned about synteny and evolution but I have not seen enough related to evolution in the result and discussion.
Response:
- The authors agree with the reviewer’s opinion. The phrase on “evolution” has been deleted from the Abstract in the revised manuscript.
Reviewer comment:
- What is the status of gene duplication in F-box gene family?
Response:
- Wheat is allohexaploid (2n = 6x = 42; AABBDD) with a huge genome of approximately 17 Gbp that consists of three closely related homoeologous subgenomes. Therefore, it is difficult to define a duplicated gene in a polyploidy genome. Therefore, homoeologous F-box genes were compared with selected F-box genes in this study in reference to the homoeologous genes provided by IWGSC Reference Sequence (v1.0). The following text, including information on homoeologous F-box genes and synteny of F-box genes, has been added to the revised manuscript:
- High-resolution and comprehensive analysis of the wheat genome by the IWGSC revealed a total of 88,733 high-confidence homologous genes of four major types (1:1:1, 1:1:0, 0:1:1, and 1:0:1) among the A, B, and D subgenomes of wheat [20]. Homoeologous F-box genes were analyzed with the information of IWGSC Reference Sequence v1.0 and possessing functional domains based on this study. A total of 921 types of homoeologous F-box genes were detected in the wheat genome. Among them, the 1:1:1, 1:1:0, 0:1:1, and 1:0:1 homoeologous groups of F-box genes were detected 332, 107, 179 and 120 times, respectively. The other minor types of homoeologous F-box genes, such as N:1:1, 1:N:1. 1:1:N, and N:0:1 were detected 213 times (Table S5). In this study, 1,311 wheat F-box gene pairs that were matched with the BLASTP algorithm with an E-value of <1e−10, minimum sequence identity of >90%, and bitscore of >500 were used to create a synteny map (Figure 5A). Among them, 558 pairs of F-box genes were detected in two subgenomes, and 510 pairs of F-box genes were identified as 255 triplets in three loci of the wheat genome. Furthermore, 243 pairs of F-box genes had more than three loci (quadruplet) in the wheat genome (Figure 5A and Table S6). (Line 236-248)
Reviewer comment:
- In the phylogeny, is different groups have some characteristics for example with respect to domain presence and expression pattern? What can be possible explanation for their functions based on these results?
Response:
- Figure 3 in the revised manuscript includes the functional domains of the F-box genes. The most abundant FBX group (1,370 genes) containing only the FBX domain at the N-terminal was excluded from the phylogenetic analysis. Section 3.3 and legend of Figure 3 have been revised as follows.
- Phylogenetic analysis was conducted using the full-length protein sequences encoded by 1,796 F-box genes (Supplementary Figure 1). Because the FBX group containing only one FBX domain in the N-terminal was the most abundant group (1,370 genes, 76% of F-box genes ofwheat) and showed high sequence diversity, phylogenetic analysis of F-box proteins possessed one or more functional domains besides FBX subgroup were showed in Figure 3. All F-box proteins containing DUF295, Cupin_8, Actin, and Arm domains were grouped together, whereas the F-box proteins possessing FBA, FBD, Kelch, and PP2 had scattered patterns within the F-box subfamilies and were grouped by proteins with other domains. The FBO subfamilies with less than five members were clustered together in one subgroup..(Line 204-212)
- Figure 3. Phylogenic analysis of wheat F-box genes. A phylogenetic tree was generated using the iTOL tool with the amino acid sequences of F-box proteins that contained FBX domains in the N-terminal and one or more known functional domains in the C-terminal of the peptide sequences.

Round 2
Reviewer 1 Report
In general, I see a big improvement in the manuscript, and thank you for making the response clear and easy to follow.
I have some further comments.
Firstly, the quality of figure 4 is still poor. It is not possible to see, at this scale, the writing or the colours of the lines that indicate the position of the genes.
Secondly, I have an issue with the methodology in the RNA extractions. I know from personal experience that extracting clean RNA from starch-heavy tissue such as wheat grains > 10 days post anthesis is difficult, and not possible with a straightforward trizol extraction. I would like to know mroe detail about the quality of your RNA extractions (nanodrop readings of possible), as I am concerned that the starch contamination would impair the quality of the sequencing.
Thirdly, there are still some issues with the english, and ofter there are spaces missing. Please proofread carefully.
Author Response
In general, I see a big improvement in the manuscript, and thank you for making the response clear and easy to follow.
I have some further comments.
Reviewer comment:
Firstly, the quality of figure 4 is still poor. It is not possible to see, at this scale, the writing or the colours of the lines that indicate the position of the genes.
Answer:
The authors made and uploaded all figures in this manuscript with high resolution (300 dpi) that can indicate the scale and the position of the genes. However, after submission process, the resolution of figures of all figures in the manuscript changed automatically to low and poor quality (see below, RC figure 1). Therefore, we have uploaded all figures with high resolution and attached original independent files separately to “supplementary section” for confirmation of figure resolution.
RC figure 1. (A) Figure 4(A) with high resolution (300 dpi) before submission step. (B) low and poor image after submission step.
Reviewer comment:
Secondly, I have an issue with the methodology in the RNA extractions. I know from personal experience that extracting clean RNA from starch-heavy tissue such as wheat grains > 10 days post anthesis is difficult, and not possible with a straightforward trizol extraction. I would like to know more detail about the quality of your RNA extractions (nanodrop readings of possible), as I am concerned that the starch contamination would impair the quality of the sequencing.
Answer:
Total RNA extraction from developing seeds was performed according to Meng (2010). In addition, the QC results of RNA for RNA sequencing are presenting in RC Table 1. The followed paragraph has been inserted in the revised manuscript (Line 133)
Total RNA of leaves was extracted from Stage 1 to 4 each developmental stage using TRIzol reagent (Invitrogen, Waltham, MA, USA) and treated with DNase I to eliminate any contaminating genomic DNA. Total RNA extraction of developing seed from Stage 5 to 7 was performed according to Meng et al. [31], for effective inhibition of RNase activity and separation from the polysaccharides to extract maximum RNA solubility.
RC Table 1. RNA QC Result
|
Sample |
ng/µl |
OD260/280 |
OD260/230 |
Total (ug) |
Ratio(28s/18s) |
RIN |
|
Stage1_1 |
2636.7 |
26.367 |
2.03 |
2.03 |
1.0 |
7.3 |
|
Stage1_2 |
2981 |
29.81 |
2.06 |
2.71 |
1.5 |
8 |
|
Stage1_3 |
2452.8 |
24.528 |
2.07 |
2.94 |
1.5 |
8 |
|
Stage2_1 |
3273.4 |
32.734 |
2.02 |
2.02 |
1.0 |
7.5 |
|
Stage2_2 |
3000.4 |
30.004 |
2.06 |
2.69 |
1.0 |
7.5 |
|
Stage2_3 |
3481 |
34.81 |
2.04 |
2.23 |
1.0 |
7.3 |
|
Stage3_1 |
2044.4 |
20.444 |
1.93 |
2.74 |
1.0 |
7.2 |
|
Stage3_2 |
1299.9 |
12.999 |
2.03 |
2.42 |
1.0 |
7.3 |
|
Stage3_3 |
1186.3 |
11.863 |
2.01 |
2.24 |
1.0 |
7.3 |
|
Stage4_1 |
1203 |
1.85 |
2.44 |
14.436 |
1.3 |
7.9 |
|
Stage4_2 |
1446.4 |
1.86 |
2.43 |
17.357 |
1.3 |
8.3 |
|
Stage4_3 |
1201.5 |
1.86 |
2.43 |
14.418 |
1.3 |
8.1 |
|
Stage5_1 |
1331.4 |
1.84 |
2.43 |
15.977 |
1.3 |
8.3 |
|
Stage5_2 |
1212.3 |
1.83 |
2.4 |
14.548 |
1.4 |
8.5 |
|
Stage5_3 |
1430 |
1.85 |
2.44 |
17.16 |
1.3 |
8.3 |
|
Stage6_1 |
640.1 |
1.82 |
2.47 |
7.681 |
1.6 |
8.8 |
|
Stage6_2 |
619.5 |
1.82 |
2.48 |
7.434 |
1.4 |
8.5 |
|
Stage6_3 |
644.8 |
1.78 |
2.52 |
7.738 |
1.5 |
8.8 |
|
Stage7_1 |
626.2 |
1.83 |
2.49 |
7.514 |
1.6 |
8.8 |
|
Stage7_2 |
630.9 |
1.8 |
2.52 |
7.571 |
1.6 |
8.9 |
|
Stage7_3 |
641.2 |
1.84 |
2.56 |
7.694 |
1.6 |
8.8 |
Reviewer comment:
Thirdly, there are still some issues with the english, and ofter there are spaces missing. Please proofread carefully.
Answer:
The English or space issues in the manuscript have been checked and revised throughout the manuscript. thank you for your kind comments
